# A comprehensive method for the quantification of medication error probability based on fuzzy SLIM

Fakhradin Ghasemi[1,2], Mohammad Babamiri[3], Zahra Pashootan[2]*

**1** Department of Occupational Health and Safety Engineering, Abadan University of Medical Sciences, Abadan, Iran, **2** Department of Ergonomics, Occupational Health & Safety Research Center, School of Public Health, Hamadan University of Medical Sciences, Hamadan, Iran, **3** Department of Ergonomics, Research Center for Health Sciences, School of Public Health, Hamadan University of Medical Sciences, Hamadan, Iran

* zahra.pashootan@gmail.com

## Abstract

Medication errors can endanger the health and safety of patients and need to be managed appropriately. This study aimed at developing a new and comprehensive method for estimating the probability of medication errors in hospitals. An extensive literature review was conducted to identify factors affecting medication errors. Success Likelihood Index Methodology was employed for calculating the probability of medication errors. For weighting and rating of factors, the Fuzzy multiple attributive group decision making methodology and Fuzzy analytical hierarchical process were used, respectively. A case study in an emergency department was conducted using the framework. A total number of 17 factors affecting medication error were identified. Workload, patient safety climate, and fatigue were the most important ones. The case study showed that subtasks requiring nurses to read the handwritten of other nurses and physicians are more prone to human error. As there is no specific method for assessing the risk of medication errors, the framework developed in this study can be very useful in this regard. The developed technique was very easy to administer.

## Introduction

Human error is one of the main causes of accidents in different organizations and industries [1]. Medical error, as a subcategory of human error, refers to any type of error which may occur during diagnosis or treatment of a disease and is estimated to lead to 44,000 to 98,000 deaths each year [2]. According to a study, 34 percent of patients in the United States have experienced medical errors, such as inappropriate medication, mistreatment, and inaccuracies in medical test results. The percentage was reported to be 30% in Canada, 27% in Australia, 25% in New Zealand, 23% in Germany, and 22% in the United Kingdom [3].

One of the most common forms of medical error is medication error. Medication errors are those medical errors that occur during the treatment process and can endanger the health

**Data Availability Statement:** All relevant data are within the paper and its Supporting Information file.

**Funding:** zahra pashootan received a fund from Hamadan University of Medical Sciences (Grant

Number: 9710045903). The funders had no role in study design, data collection and analysis, decision to publish, or preparation of the manuscript.

**Competing interests:** Authors declare no conflict of interest.

and safety of patients [4]. Medication itself is a term referring to "any product containing compounds with a proven biological effect" [4]. Medication errors have been reported to be associated with undesired consequences such as mortality, increased length of stay, and increased costs of treatment and hospitalization [5]. Each year, 100,000 people in the United States lose their lives because of medical errors, out of which, 7000 cases are associated with medication errors [6]. Moreover, in a UK survey in 2018, it was revealed that more than 2 million people suffer from medication-related injuries annually, while nearly 100,000 ones lose their lives as a result of such injuries. According to previous studies, the cost of medication-related adverse events in 2018 is estimated to be approximately $ 300 million [7].

Among hospital staff, nurses play an important role in the treatment process, therefore the rate of medication error among them is high [8]. Various factors can contribute to medication errors among nurses. Nurses believe that factors such as the use of short names instead of full medication names, similarity in medication names, carelessness of nurses, high work pressure and workload, particularly during emergencies [5, 9, 10], low medication information, and weaknesses in continuous education and training are the main causes of medication error [11–14]. An important strategy in preventing medication errors is error reporting [15]. But the error reporting process is not implemented well, and nurses refuse to report the error because they fear the consequences and worry about losing their jobs [16, 17]. In developing countries, statistics are not available due to inaccurate reporting of medication errors [5].

Many techniques have been developed for assessing human reliability and predicting human error probability (HEP), including the Human Error Assessment & Reduction Technique [18], the Standardized Plant Analysis Risk [19], The System Human Error Reduction and Prediction Approach [20], and the Cognitive Reliability and Error Analysis Method [21]. These techniques attempt to evaluate human reliability in different fields. However, none of them is developed to be specifically used in hospitals for medication error assessments. The factors affecting human performance are different from one domain to another. In power plant industries, it is believed that factors such as adherence to safety principles, age, and training have the greatest impact on human error [22]. In repair and maintenance tasks of the aviation industry, factors such as unfavorable physiological conditions, physical and mental limitations, coordination, communication, and planning [23] and in the oil and gas industry, defects in the design of equipment and defects in management systems are the most important factors affecting human error [24]. In the maritime transport industry, stress, experience, complexity, training, time availability, environmental factors, safety culture, and communication are important factors influencing human error [25]. As factors affecting human error probability differ from one domain to another, it is of pivotal importance to develop a technique specifically for the quantification of human error probability in medication processes.

Uncertain and ambiguous data and opaque experts' opinions are other issues in human reliability analysis (HRA). Such data and knowledge can make the results of HRA unreliable. Fuzzy logic theory is specialized for dealing with this issue [26]. In contrast to the Boolean logic, the membership degree of a variable to a set can be any real value between zero and one in the Fuzzy environment. Imprecise and vague information obtained in the shape of linguistic terms from domain experts can be manipulated, processed, and represented mathematically using the Fuzzy sets theory and associated Fuzzy operators. Accordingly, Fuzzy logic has found many applications in HRA. For example, techniques such as CREAM, HEART, and SHERPA have been equipped with Fuzzy logic in different ways to reduce subjectivity and enhance the reliability of results [27–30]. Consequently, it seems rational, useful, and even necessary to employ Fuzzy logic in developing any new HRA technique.

Therefore, the aim of the present study was to develop a new method for assessing the medication error probability. The proposed method had the following advantages:

- In this method, all factors supposed to affect medication errors are included,

- The method is based on the principles of fuzzy logic which is the preferred approach in dealing with uncertain and ambiguous data,

- The method is based on a well-accepted foundation methodology for human error analysis,

- The method is specialized to assess the probability of medication errors.

## Methods

In order to develop a specific and comprehensive method for estimating the medication error probability, first, an extensive literature review was conducted to explore factors supposed to affect such errors. Second, these factors were categorized into three main groups. These groups were called Performance Shaping Factors (PSFs) and factors under each group were called sub-Performance Shaping Factors (subPSFs). Third, the weight of these PSFs and subPSFs was calculated using fuzzy multi-criteria decision-making methods. Lastly, the Success Likelihood Index Methodology (SLIM) was employed to build a function based on PSFs and subPSFs for estimating human error probability during medication processes. Fig 1 demonstrates the study procedures and main steps carried out to conduct this study. Moreover, this study was reviewed and approved by the ethics committee of Hamadan University of medical sciences. The ethics code assigned to the study was: IR.UMSHA.REC.1397.679. In the following sections, the mentioned steps were described in detail.

**Phase 1**
Identifying all factors affecting medication error probability (subPSFs):

**Activities**
1- Literature review,
2-Investigating available human reliability assessment techniques,

**Phase 2**
Categorizing subPSFs into three mains PSFs:

**Activities**
- categorizing all identified subPSFs into three main categories of organization, job, and person-related factors based on a model from Health and Safety Executive (HSE).

**Phase 3**
Calculating the weight of PSFs using Fuzzy MADM:

**Activities**
1- Gathering opinions of experts,

2- Calculating the relative weight of experts,

3- Calculating the degree of agreement (similarity) between each pair of experts,

4- Calculating the Average of Agreement for each expert,

5- Calculating the Relative Agreement for each expert,

6- Calculating the consensus degree coefficient for each expert,

7- Aggregating opinions of experts,

8- Defuzzification and calculating the weight of subPSFs,

**Phase 5**
Developing Fuzzy SLIM method and calculating human error probability:

**Activities**
1- Developing the Fuzzy SLIM method,

2- Hierarchical Task Analysis (HTA) of an emergency department,

3- Determining the ratings of subPSFs,

4- Calculating human error probability using the Fuzzy SLIM method

**Phase 4**
Calculating the weight of subPSFs using Fuzzy AHP:

**Activities**
1- Gathering opinions of experts,

2- Calculation of the fuzzy synthetic extent values with respect to the ith criterion (Ci),

3- Determination of the degree of possibility of M1 (l1, m1, u1) ≥ M2 (l2, m2, u2),

4- Calculation of the degree of possibility for a convex fuzzy number to be higher than k convex fuzzy numbers,

5- Normalization of the weight vector,

**Fig 1. Overview of the methodology used to quantify the medication error probability.**

## Determination of factors affecting medication errors

Factors affecting medication errors were identified by reviewing papers indexed in three databases: Web of Science, Scopus, and PubMed. Moreover, all performance shaping factors used in popular human reliability assessment techniques such as CREAM, SPAR-H, and HEPI, were also taken into consideration. As the number of these factors was high, it may be difficult and very challenging to include all of them as PSFs in a human reliability assessment technique. Therefore, the authors decided to categorize them into three main groups: organization-related factors, job-related factors, and personal factors.

## Fuzzy SLIM method

For determining the probability of medication error based on the factors identified in the previous step, an equation was developed based on fuzzy sets theory and SLIM) SLIM was first developed by Embrey et al. in 1984 [31]. The methodology benefits from a strong foundation and has been extensively used for human reliability assessment. It is based on the fact that the probability of an error occurring in a particular situation is a function of PSFs. SLIM utilizes the following equations for calculating human error probability:

$$SLI = \sum\nolimits_{i=1}^{n} R_i . W_i, \qquad 1 \leq SLI \leq 9 \tag{1}$$

$$Log(HEP) = a.SLI + b \tag{2}$$

Where SLI stands for Success Likelihood Index, $R_i$ stands for Rating of $PSF_i$, Wi stands for Weighting of $PSF_i$, HEP stands for Human Error Probability, and a and b are two constants that should be calculated based on the minimum and maximum human error probabilities.

For developing the SLI equation, two steps should be performed. In the first step, called PSFs weighting, the weight of PSFs should be determined. In the second step, called PSFs rating, PSFs should be rated based on the real situation of the organization. In the following sections, the methods based on which these steps should be conducted are explained.

**PSFs weighting.** Traditionally, direct opinions of the experts have been used for weighting PSFs in SLIM. This approach has been regarded as a weakness because of the lack of a scientific basis and uncertainty relating to human judgment [25]. In this study, for covering this weakness, the fuzzy multiple attributive group decision-making methodology (FMAGDM) developed by Ölçer and Odabaşi [32] was used. This method was also used by Akyuz [25] for the weighting of PSFs in developing the SLIM equation. The main advantage of this method is the use of fuzzy logic in collecting and integrating experts' opinions. Fuzzy logic is especially helpful in dealing with vague and uncertain information [26]. Accordingly, most multi-criteria decision-making methods have been upgraded based on Fuzzy logic and Fuzzy sets theory [33]. Using Fuzzy logic, the opinions of the experts are gathered in the form of linguistic terms instead of numerical values. Apparently, expressing opinions in terms of linguistic terms is more compatible with the human information processing system and people are more comfortable to express their opinions in terms of linguistic terms instead of crisp values. Moreover, the methodology used in this study benefits from the similarity aggregation method (SAM) for integrating opinions from different experts. SAM is a well-accepted method for making a consensus among opinions gathered from different experts [34]. The method employs a coefficient, known as β, in the aggregation step that provides the analyst or whoever conducts HRA with more dominance and authority over the problem [32]. Moreover, Akyuz [25] used this approach for calculating the weight of PSFs in the SLIM equation.

In this step, the opinions from ten experts, including five occupational health and safety experts and five nurses, were collected. Participation of experts was voluntary and they signed an informed consent form.

The following steps were carried out to determine the weights of PSFs:

- **Step 1:** In this step, the experts were asked to express their opinions regarding the importance of a PSF in developing human error. For example, they were asked: How important are "the organizational factors" in developing human error? They should answer this question using linguistic terms. Depicted in Fig 2 are the linguistic terms and associated fuzzy numbers used in this study.

- **Step 2:** In this step, the relative weight of each expert was determined. Work experience, education level, and relevancy to the field of study were factors based on which the relative weight of experts was calculated.

- **Step 3:** in this step, the degree of agreement (similarity) of the opinions between each pair of experts was calculated using Eq 3:

$$S_{uv}\left(\widetilde{R}_u, \widetilde{R}_v\right) = 1 - \frac{1}{J = 3 \text{ or } 4} \sum_{i=1}^{3 \text{ or } 4} |a_i - b_i| \tag{3}$$

Where $S_{uv}\left(\widetilde{R}_u, \widetilde{R}_v\right)$ is the degree of agreement between experts u and v, $a_i$ and $b_i$ are the corresponding members of A and B respectively and J is equal to 3 and 4 for triangular and trapezoidal fuzzy numbers, respectively.

- **Step 4:** The Average of Agreement for each expert, $AA(E_u)$, was calculated using Eq 4:

$$AA(E_u) = \frac{1}{m-1} \sum_{u \neq v}^{m} S_{uv}\left(\widetilde{R}_u, \widetilde{R}_v\right) \tag{4}$$

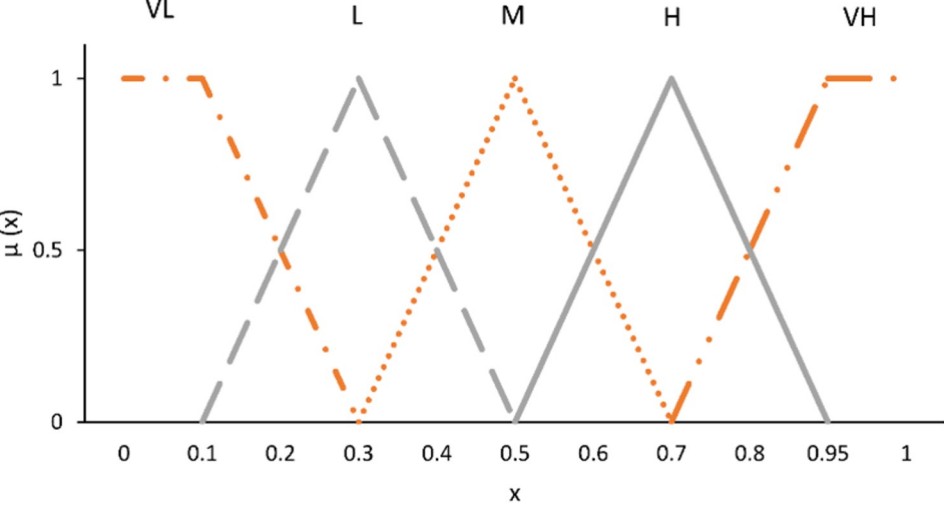

**Fig 2. The linguistic terms used for collecting experts' opinions.**

- **Step 5:** The Relative Agreement for each expert, $RA(E_u)$ was computed using Eq 5:

$$RA(E_u) = \frac{AA(E_u)}{\sum_{u=1}^{m} AA(E_u)} \tag{5}$$

- **Step 6:** In this step, the consensus degree coefficient for each expert, $CC(E_u)$, was calculated using Eq 6:

$$CC(E_u) = \beta.W(E_u) + (1 - \beta).RA(E_u) \tag{6}$$

Where $W(E_u)$ is the weight of expert U calculated in step 2 and $\beta$ is a factor used to indicate the importance of W over RA. It can have a value ranging from zero to one. If all experts have the same degree of importance, then $\beta$ should be taken as zero. In this study, the value of $\beta$ was considered to be 0.4.

- **Step 7:** In this step, the fuzzy opinions from all experts were aggregated using Eq 7:

$$R_{AG} = \sum_{i=1}^{n} CC(E_i).R(E_i) \tag{7}$$

Where $R(E_i)$ is the opinion of expert i in terms of fuzzy sets.

- **Step 8:** In this step, $R_{AG}$ calculated in the previous step was defuzzified. Defuzzification is a process through which a fuzzy number is transformed into a crisp value. There are several ways for defuzzification. In this study, the max-min method developed by Chen and Hwang [35] was applied.

$$\mu_{max}(x) = \begin{cases} x & \text{for } 0 \le x \le 1 \\ 0 & \text{otherwise} \end{cases} \tag{8}$$

$$\mu_{min}(x) = \begin{cases} 1 - x & \text{for } 0 \le x \le 1 \\ 0 & \text{otherwise} \end{cases} \tag{9}$$

Next, the right and left scores of the fuzzy number (B) are calculated using Eqs 10 and 11, respectively:

$$\mu_R(B) = \sup_x \left[ \mu_B(x) \wedge \mu_{max}(x) \right] \tag{10}$$

$$\mu_L(B) = \sup_x \left[ \mu_B(x) \wedge \mu_{min}(x) \right] \tag{11}$$

Since the right and left scores are calculated, the total score is calculated using Eq 12:

$$\mu_T(B) = \left[ \mu_R(B) + 1 - \mu_L(B) \right]/2 \tag{12}$$

**PSFs rating.** As mentioned before, there were three PSFs in this study: organizational factors, job factors, and personal factors, and each of these PSFs was a set of subPSFs affecting human reliability. Therefore, for rating these PSFs, we should first determine the rating of the associated subPSFs. However, these PSFs are different in terms of importance and ability in affecting human reliability. Consequently, the weight of PSFs should also be computed. To do this, Fuzzy AHP method developed by Chang [36] was employed. The reason behind using fuzzy AHP in this phase of the study was its simplicity and also encompassing the advantages of both AHP and Fuzzy logic [37, 38]. Moreover, there were only a limited number of subPSFs under each PSF, so the number of required pair-wise comparisons was rationally low. The steps of fuzzy AHP were as follows:

- **Step 1:** Development of a fuzzy pair-wise comparison matrix. The linguistic terms used by experts for making comparisons between each pair of subPSFs are presented in Table 1.

- **Step 2:** Calculation of the fuzzy synthetic extent values with respect to the ith criterion (Ci). Eq 13 was used for this purpose:

$$S_{ci} = \sum_{j=1}^{m} M_{gi}^{j} \odot \left[ \sum_{i=1}^{n} \sum_{j=1}^{m} M_{gi}^{j} \right]^{-1} \tag{13}$$

- **Step 3:** Determination of the degree of possibility of $M_1$ $(l_1, m_1, u_1) \geq M_2$ $(l_2, m_2, u_2)$.

$$V(M_1 \geq M_2) = \sup_{x \geq y} \left[ \min \left( \mu_{M_1}(x) . \mu_{M_2}(y) \right) \right] \tag{14}$$

The computation of both $V(M_1 \geq M_2)$ and $V(M_2 \geq M_1)$ are required for comparing $M_1$ and $M_2$. The following equations were used in this step:

$$V(M_1 \geq M_2) = 1 \text{ if } m_1 \geq m_2 \tag{15}$$

$$V(M_2 \geq M_1) = hgt(M_1 \cap M_2) = \frac{l_1 - u_2}{(m_2 - u_2) - (m_1 - l_1)} \tag{16}$$

- **Step 4:** Calculation of the degree of possibility for a convex fuzzy number to be higher than k convex fuzzy numbers, i.e., $M_i$ (i = 1, 2, 3, . . ., k). Eq 17 is used in this step:

$$V(M \geq M_1.M_2.M_3. \ldots .M_k) = V[(M \geq M_1) \text{ and } (M \geq M_2) \text{ and} \ldots \text{and } (M \geq M_k)]$$
$$= \min V(M \geq M_i).i = 1.2.3. \ldots .k \tag{17}$$

**Table 1. Linguistic terms and corresponding fuzzy numbers used in fuzzy AHP.**

| Linguistic term | Weight definition | Fuzzy number |
|---|---|---|
| Equal importance | 1 | (1, 1, 1) |
| A little more importance | 2 | (1, 1.5, 2) |
| Relatively more importance | 3 | (1.5, 2, 2.5) |
| Much more importance | 4 | (2, 2.5, 3) |
| Very much more importance | 5 | (2.5, 3, 3.5) |

Next, the weight vector is determined:

$$W' = \left(W_1'. \, W_2'. \, W_3'. \ldots . W_n'\right)^{T} \tag{18}$$

Where for $W_i' = \min V(S_i \geq S_k)$ for $k = 1.2.3. \ldots . n; \ k \neq i$

- **Step 5:** normalization of the weight vector $w = (w_1, w_2, w_3, \ldots, w_n)^{T}$ using Eq 19;

$$W_i = \frac{W_i'}{\sum_{i=1}^{n} W_i'} \tag{19}$$

After determining the weight of subPSFs, the rate of each PSFs category was determined using Eq 20:

$$R = \sum r_i . w_i \tag{20}$$

Where $R$ = rating of the category, $r_i$ = rating of a subPSFs, and $w_i$ is the weight of a subPSFs.

The weight of each factor can be in a range from zero to one and the sum of weights of factors in each category should be equal to one. The opinions of the same panel of experts were used in this step. Rating of all subPSFs should be determined by consulting the staff of the organization. This rating can range from 1 (the worst situation) to 9 (the best situation).

## Results

### Factors affecting medications errors

As mentioned previously, an extensive literature review was conducted to extract factors affecting medication errors. A total number of 17 factors was identified in this step. These factors alongside their definitions are presented in Table 2. Because the number of these factors was high, we categorized them into three categories: personal, job, and organization-related factors. These three categories of variables were called PSFs and factors under each category were called subPSFs.

### The Fuzzy SLIM equation

According to FMADM, the weights of personal, job, and organization-related factors in the SLI equation were equal to 0.367, 0.311, and 0.321, respectively. Therefore, the SLI can be calculated using the following equation:

$$SLI = (0.367 \times R_{per}) + (0.311 \times R_{job}) + (0.321 \times R_{org}) \tag{21}$$

Where $R_{per}$ is the rating of PSF personal, $R_{job}$ is the rating of PSF job, and $R_{org}$ is the rating of organization-related factors. Moreover, for calculating the rating of each PSF, the weight of each subPSF was calculated separately using Fuzzy AHP. The results are presented in Table 3. Accordingly, among subPSFs related to the job category, workload had the highest weights while the physical environment had the lowest weight. Among the subPSFs in the organization category, the highest and the lowest weights were associated with patient safety climate and communication between staff, respectively. Likewise, among the subPSFs in the personal category, fatigue and physical health of the nurse had the highest and the lowest weights, respectively.

For determining a and b in Eq 2, the lowest and highest human error probabilities were considered to be 0.0001 and 1, respectively.

**Table 2. Factors affecting medication errors and their definitions based on a literature review performed in various databases.**

| Factor (subPSF) | Definition | Category (PSF) |
|---|---|---|
| Knowledge | Knowledge is information and understanding of the subject that one has, or that all people possess [39]. | Personal |
| Experience | Work-related knowledge and knowledge over the years [25]. | |
| Fatigue | Fatigue is a psychological aspect of not having enough energy to do the job and not having the mental drive to continue a job [40]. | |
| Physical health | Conditions where the body is free from any disease, abnormal, and in favorable conditions [39]. | |
| Task Time (Circadian Rhythm) | When the task is done at that time [41]. | |
| Workload | The relationship between one's mental processing ability or resources with the amount of work required of the individual [42]. | Job |
| Procedures | Who, what to do, when, and under what criteria [39]. | |
| The physical environment | Factors affecting staff performance such as weather, hospital environment, nursing station conditions, and medication store conditions [42]. | |
| Housekeeping | Inadequate and inappropriate physical conditions (crowded work environment, telephone, space constraint, noise, patient companions around) [43]. | |
| Transparency of responsibilities | The specificity of each person's task for that person is clear | |
| Time available | The time frame in which employees have to perform their task in an abnormal event [25]. | |
| Patient safety climate | A common understanding among group members about the methods, practices, and types of behaviors that are rewarded and supported with regard to patient safety [44]. | Organization |
| Safety culture | The result of individual and group beliefs, values, attitudes, perceptions, competencies, and behavior patterns that determine an organization's commitment to patient quality and safety. A set of what is being pursued in the organization for safe healthcare [45]. | |
| Training | Systematically develop the knowledge, skills, and attitudes needed to perform a specific task [46]. | |
| Communication between staff | The process of transferring information and understanding from one person to another [47]. | |
| Supervising staff | Planning, organizing, directing, and controlling work and employee activities [48]. | |
| Error Management Culture | It is an approach that does not attempt to fix the errors completely but attempts to deal with and communicate the errors and their consequences after the error has occurred [49]. | |

## Case study

A case study was conducted in the emergency department of a hospital located in Hamadan, Iran. First, a task analysis was performed by interviewing an experienced nurse. The results are presented in Table 4. As evident, five tasks and 31 sub-tasks are required for admission and treatment of a patient in an emergency department.

Next, the analyst visited the emergency department and interviewed nurses. Based on the direct observation and information obtained from nurses, the rating of subPSFs was determined. Each subPSF can have a value within a range from 1 to 9 so that higher values are indicative of a better situation of that subPSF. SLI and then HEP were calculated based on the rating of subPSFs.

HEP for each subtask is presented in Table 5. Accordingly, subtask 2.4, "writing medications information (including name, dosage, prescription and prescription time) in the patient's Kardex based on the information recorded in the PRF", had the highest HEP, and subtask 3.1, "receiving the prescribed medications from the hospital drugstore by an assistant", and subtask 3.2, "receiving medications from the assistant by a nurse", had the lowest HEP.

## Discussion

Medication errors endanger the health and safety of patients. Medication errors can be affected by a wide range of organization, job, and personal factors. There is no universal

**Table 3. Weight of job sub-variable.**

| SubPSFs under the job PSF | | Weight |
|---|---|---|
| | Workload | 0.385 |
| | Availability of work procedures | 0.175 |
| | The physical environment | 0.039 |
| | Housekeeping | 0.135 |
| | Transparency of responsibilities | 0.110 |
| | Time available | 0.156 |
| | **Sum** | 1 |
| **SubPSFs under the organization PSF** | | |
| | Patient safety climate | 0.225 |
| | Safety culture | 0.209 |
| | Training | 0.210 |
| | Communication between staff | 0.099 |
| | Supervising staff | 0.120 |
| | Error Management Culture | 0.136 |
| | **Sum** | 1 |
| **SubPSFs under the personal PSF** | | |
| | Knowledge | 0.237 |
| | Experience | 0.270 |
| | Fatigue | 0.313 |
| | Physical health of the nurse | 0.065 |
| | Task time (Circadian Rhythm) | 0.114 |
| | **Sum** | 1 |

method to identify possible errors during medication processes and determine the associated risks. Therefore, the present study aimed to develop a framework for calculating the HEPs during the medication process. In this study, an extensive literature review was conducted in order to identify factors affecting human error in healthcare. Next, all those factors relevant to medication processes were extracted and categorized into three main groups; organizational, job, and personal factors. A fuzzy SLIM approach was developed to predict HEP based on these three groups of factors. The procedure used in this study can be applied to any other domain but all phases, including identifying subPSFs, weighting them, and rating them, must be repeated.

The method developed in this study can evaluate the combined effect of seventeen factors on medication error probability, making it more comprehensive than available HRA techniques. Some of these factors such as workload, availability of work procedures, and safety culture are general to any occupational setting, but some others such as patient safety climate, physical health of nurses, and error management culture are unique to healthcare environments. Taking into account these variables is an important advantage of this method over other available techniques. Moreover, the weights calculated in this study are specific to healthcare environments. For example, among organizational factors, patient safety climate had the highest weight, meaning that it is the most important organizational factor affecting the medication error probability. Such a factor with its relative importance has not been considered in any available HRA technique.

Among the job-related factors, workload had the highest weight. The effect of workload on HEP has been demonstrated in previous studies [50–52]. Workload can increase the probability of human error in both direct and indirect ways. The probability of attention failures

**Table 4. Task analysis in an emergency department.**

| Subtask |
| --- |
| 1. Admitting patient and recording his/her information |
| l.1. Recording patient demographic information on all pages of the patient records file (PRF) |
| 1.2. Registration of patient bed number in the PRF |
| 1.3. Writing the patient's medical history in the PRF |
| 1.4. Placing the identification wristband on the patients' wrist |
| 2. Examination and registration of medications required for the patient |
| 2.1. Initial examination of the patient by physician and nurse and recording the detailed medication process |
| 2.2. Striking through the medications that no longer need to be prescribed (If there are) by the nurse using a red pen and writing the "D.C" word in the front of it, |
| 2.3. Recording information about new medications in the PRF, |
| 2.4. Writing medications information (including name, dosage, prescription, and prescription time) in the patient's Kardex based on the information recorded in the PRF |
| 2.5. Clearing out old medicines from the patient card, |
| 2.6. Writing medication information (name, dosage, prescription instructions, and prescription time) in the patient's card based on the information recorded in the PRF using a pencil, |
| 2.7. Registration of the required medications in the HIS system |
| 3. Receiving and storing medications |
| 3.1. Receiving the prescribed medications from the hospital drugstore by an assistant, |
| 3.2. Receiving medications from assistant by a nurse |
| 3.3. Initial review of medications by the nurse for the detection of any discrepancy, |
| 3.4. Separating sensitive medications from non-sensitive ones (sensitive medications are those that require non-routine instructions of prescribing). |
| 3.5. Labeling sensitive medications (prescription protocol) |
| 3.6. Separating refrigerating medications from non- refrigerating, |
| 3.7. Putting the medications in the medication room and designated shelves, |
| 4. Preparation of medications |
| 4.1. Finding the patient's medication card |
| 4.2. Reading medication information from the medication card |
| 4.3. Finding medications in the medication room, |
| 4.4. Picking medications from the designated shelves, |
| 4.5. Checking medication name, prescription instructions, and the expiry date |
| 4.6. Obtaining the desired dosage of the medication and preparing it if required, |
| 4.7. Transferring the prepared medications to the emergency ward in special trays, |
| 5. Prescribing medication to the patient |
| 5.1. Identifying the intended patient in the emergency ward, |
| 5.2. Asking the patient her/his name |
| 5.3. Matching the patient name with her/his wristband, |
| 5.4. Check the prescription medication guidelines and determine the prescription medication route |
| 5.5. Prescribing medications under their instructions |
| 5.6. Checking patients for probable side effects of prescribed medications up to 15 minutes |

increases as workload rises [53]. Moreover, high workload can lead to an increased level of occupational fatigue [54], thereby raising the probability of human error and unsafe behaviors. Availability of work procedures was another high-rated job-related factor. The lack of up-to-date and easy-to-follow work procedures is a main cause of human error in various fields. Failure to adhere to work procedures is a leading cause of medication errors [55].

Among the organization related factors, the highest weight was associated with patient safety climate. A study demonstrated that a positive safety climate is associated with lower

**Table 5. The human error probability in each subtask alongside the corresponding PSFs rating.**

| Task/Subtask | $R_P$ | $R_J$ | $R_O$ | HEP |
|---|---|---|---|---|
| 1.1 | 5.814 | 4.49 | 3.593 | 1.44E-2 |
| 1.2 | 5.814 | 4.49 | 4.28 | 1.12E-2 |
| 1.3 | 5.814 | 4.49 | 3.956 | 1.25E-2 |
| 1.4 | 6.384 | 4.49 | 4.604 | 0.78E-2 |
| 2.1 | 6.042 | 5.08 | 4.11 | 0.88E-2 |
| 2.2 | 6.156 | 3.925 | 3.566 | 1.54E-2 |
| 2.3 | 5.928 | 3.925 | 4.324 | 1.28E-2 |
| 2.4 | 5.928 | 3.925 | 3.033 | 2.06E-2 |
| 2.5 | 6.27 | 3.925 | 3.242 | 1.65E-2 |
| 2.6 | 5.814 | 4.275 | 3.132 | 1.84E-2 |
| 2.7 | 6.042 | 4.275 | 3.748 | 1.33E-2 |
| 3.1 | 6.498 | 4.888 | 4.687 | 0.62E-2 |
| 3.2 | 6.498 | 4.888 | 4.687 | 0.62E-2 |
| 3.3 | 6.156 | 3.868 | 3.85 | 1.41E-2 |
| 3.4 | 6.27 | 3.868 | 4.478 | 1.07E-2 |
| 3.5 | 6.27 | 4.043 | 4.181 | 1.12E-2 |
| 3.6 | 6.27 | 3.733 | 4.687 | 1.04E-2 |
| 3.7 | 6.384 | 4.138 | 4.268 | 1.00E-2 |
| 4.1 | 6.384 | 4.248 | 4.269 | 0.96E-2 |
| 4.2 | 6.156 | 4.248 | 4.39 | 1.02E-2 |
| 4.3 | 6.384 | 4.248 | 5.106 | 0.70E-2 |
| 4.4 | 6.384 | 4.248 | 4.897 | 0.76E-2 |
| 4.5 | 6.156 | 4.248 | 3.873 | 1.23E-2 |
| 4.6 | 6.042 | 4.248 | 4.896 | 0.88E-2 |
| 4.7 | 6.384 | 3.573 | 4.896 | 0.97E-2 |
| 5.1 | 6.042 | 3.534 | 4.379 | 1.39E-2 |
| 5.2 | 6.042 | 3.669 | 4.071 | 1.47E-2 |
| 5.3 | 6.042 | 3.669 | 4.197 | 1.41E-2 |
| 5.4 | 6.042 | 3.844 | 4.896 | 1.02E-2 |
| 5.5 | 6.042 | 4.019 | 4.687 | 1.03E-2 |
| 5.6 | 6.27 | 3.669 | 3.43 | 1.69E-2 |

perceived error occurrence among nurses [56]. Error reporting is a main dimension of patient safety climate. Error communication is a pivotal dimension of error management in hospitals. Feedback and communication about errors can be an effective way of reducing HEP in hospitals. Error reporting is mandatory in many hospitals but experience shows that nurses are reluctant to report all committed errors because of fear of being blamed, which is why the effectiveness of error reporting systems in reducing the rate of medical errors has been questioned in some studies [57]. Training regarding causes and consequences of medication errors was another important factor in this category. Training has been introduced as an effective way of reducing medication errors in various studies [58, 59].

According to the results of the present study, among personal factors, fatigue was the most important one affecting HEP. This finding is in line with those of previous studies. Fatigue, as a result of workload, sleep deprivation, or long working hours, can degrade mental functions and thereby increasing the probability of human errors [54, 60]. Experience and knowledge were other important personal factors affecting HEP. Lack of knowledge regarding types of medications to be used, patient condition, and dose calculation can lead to medication errors

[61]. Björkstén et al. [62] also reported that lack of knowledge contributed in 13% of medication errors by nurses in Sweden hospitals.

For calculating HEP, an equation was developed based on the Fuzzy SLIM. SLIM is a well-founded method for human error analysis in any domain. The methodology has been used as a basis for development of a variety of human error assessment techniques. Based on SLIM, Islam et al. [63] developed a monograph human error assessment in marine operations and Akyuz [25] developed another method for human error assessment during abandon ship procedures.

In this study, we conducted a case study in a hospital emergency department. The highest HEP was associated with the subtask 2.4 where nurses transform information regarding medications from the PRF to the patient's Kardex. Understanding the handwritten orders may be difficult for inexperienced nurses, as a result the probability of error in this step is high. Moreover, any distracting agent during this subtask can lead to attention failures (slips) and memory failures (lapses). Unfortunately, in the emergency department investigated in the present study there were many sources of distraction, such as noises from patients' companions. The HEP in the subtask 2.6 was also high because of the same reasons.

Worth mentioning, the consequences of medication errors are not straightforward to be determined. The consequences can vary across different patient conditions and medication types. Therefore, the probability of medication error may be a better indicator than the risk for implementing control measures.

This study has several limitations which can be the subject of future studies. In this study, the interactions among factors affecting medication errors were not investigated. SLIM cannot capture the interrelationships among a set of PSFs, but other approaches such as Bayesian networks can do this. The weighting process should be repeated in other countries in order to make a comparison about the relative importance of factors affecting medication errors all around the world.

## Conclusion

This study developed a new tool for computing the probability of medication errors based on the Fuzzy SLIM approach. Workload, patient safety climate, and fatigue were the most important factors affecting the probability of medication errors. The case study demonstrated that the method is easy to administer. Hospital managers and occupational health and safety practitioners can use the method developed in this study to monitor and manage medication errors in hospitals.

## Supporting information

**S1 File. Data used in the study.**
(PDF)

## Acknowledgments

The authors would like to thank all experts cooperated with the authors in this study.

## Author Contributions

**Conceptualization:** Fakhradin Ghasemi, Mohammad Babamiri, Zahra Pashootan.

**Data curation:** Fakhradin Ghasemi, Mohammad Babamiri, Zahra Pashootan.

**Formal analysis:** Fakhradin Ghasemi, Mohammad Babamiri, Zahra Pashootan.

**Funding acquisition:** Zahra Pashootan.

**Investigation:** Fakhradin Ghasemi, Mohammad Babamiri, Zahra Pashootan.

**Methodology:** Fakhradin Ghasemi, Mohammad Babamiri, Zahra Pashootan.

**Project administration:** Fakhradin Ghasemi, Mohammad Babamiri.

**Resources:** Fakhradin Ghasemi, Zahra Pashootan.

**Supervision:** Fakhradin Ghasemi.

**Validation:** Fakhradin Ghasemi, Zahra Pashootan.

**Visualization:** Fakhradin Ghasemi, Zahra Pashootan.

**Writing – original draft:** Fakhradin Ghasemi, Mohammad Babamiri, Zahra Pashootan.

**Writing – review & editing:** Fakhradin Ghasemi, Zahra Pashootan.

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
