## [Decision Letter · Decision Letter 0]

10 Dec 2021

PONE-D-21-36423A comprehensive method for the quantification of medication error probability based on fuzzy SLIMPLOS ONE

Dear Dr. Pashootan,

Thank you for submitting your manuscript to PLOS ONE. After careful consideration, we feel that it has merit but does not fully meet PLOS ONE’s publication criteria as it currently stands. Therefore, we invite you to submit a revised version of the manuscript that addresses the points raised during the review process.

We look forward to receiving your revised manuscript.

Kind regards,

Yang Li

Academic Editor

PLOS ONE

Journal Requirements:

2. Please ensure that you have specified (1) whether consent was informed, (2) what type you obtained (for instance, written or verbal, and if verbal, how it was documented and witnessed). If your study included minors, state whether you obtained consent from parents or guardians. If the need for consent was waived by the ethics committee and (3) If you are reporting a retrospective study of medical records or archived samples, please ensure that you have discussed whether all data were fully anonymized before you accessed them and/or whether the IRB or ethics committee waived the requirement for informed consent. If patients provided informed written consent to have data from their medical records used in research, please include this information.

"The authors would like to thank Hamadan University of medical sciences for financial supports (Grant Number: 9710045903)."

"zahra pashootan received a fund from Hamadan University of Medical Sciences (Grant Number: 9710045903). "

Reviewers' comments:

Reviewer's Responses to Questions

**Comments to the Author**

1. Is the manuscript technically sound, and do the data support the conclusions?

Reviewer #1: Yes

Reviewer #2: Yes

2. Has the statistical analysis been performed appropriately and rigorously? 

Reviewer #1: Yes

Reviewer #2: Yes

3. Have the authors made all data underlying the findings in their manuscript fully available?

Reviewer #1: No

Reviewer #2: Yes

4. Is the manuscript presented in an intelligible fashion and written in standard English?

Reviewer #1: No

Reviewer #2: Yes

5. Review Comments to the Author

Reviewer #1: This paper presents a comprehensive method for the quantification of medication error probability based on

fuzzy SLIM. In general, the topic is interesting, but this work is not well presented. I think that the quality of this paper can be improved if the authors address the following aspects:

1 The main contributions and novelty of this paper should be further summarized. This reviewer suggests the authors exactly mention what is new compared with existing approaches and why the proposed approach is needed to be used instead of the existing methods. This reviewer suggests the authors use bullets (3 bullets is standard) and in each bullet explain one contribution clearly.

2 How scalable is the presented approach?

3 This reviewer would like to suggest the authors add a flowchart of the presented method since more details can enrich the paper.

4 The proposed method might be sensitive to the values of its main controlling parameters. How did you tune the parameters?

5 The novelty and contribution of the present work need further justification. Key results need to contrast and compare with the results of other state-of-the-art methods in the literature.

Reviewer #2: This paper is technically sound. There are still some concerns of this reviewer:

1 Author should highlight the research gaps and contribution of the proposed work by comparing the state of the art methods and recent studies.

2 Better justify the selected Fuzzy multiple attributive group decision making (MAGDM) methodology and fuzzy analytical hierarchical process (FAHP). There are so many multiple attribute decision making methods. Why did you choose these two methods, based on what evidence?

3 In the introduction section, the literature review should be strengthened. Authors are recommended to include and review some studies regarding MAGDM and FAHP applications such as [doi.org/10.1371/journal.pone.0251940], [doi.org/10.1155/2016/3965608], [doi.org/10.1371/journal.pone.0239140] to improve the literature survey:

4 Authors have not presented the limitations of this work. How this work can be extended in practical applications? Please elaborate on it.

5 The italic types of symbols in equations and the main text should be in the same expression.

6 Although the manuscript is well written in terms of English, there are some (very few, indeed) grammatical and expression errors. It is suggested to proofread the paper.

6. PLOS authors have the option to publish the peer review history of their article (what does this mean?). If published, this will include your full peer review and any attached files.

Reviewer #1: No

Reviewer #2: No

---

## [Author Response · Author response to Decision Letter 0]

7 Jan 2022

We would like to thank the reviewers for taking time to review this paper and for their useful and constructive comments which helped us to improve the quality of the paper. Meanwhile, we addressed all comments one by one. All changes are highlighted in yellow in the 'Revised Manuscript with Track Changes' file. 

Comments for the reviewer #1 and Response of the authors

1-The main contributions and novelty of this paper should be further summarized. This reviewer suggests the authors exactly mention what is new compared with existing approaches and why the proposed approach is needed to be used instead of the existing methods. This reviewer suggests the authors use bullets (3 bullets is standard) and in each bullet explain one contribution clearly. We would like to thank the reviewer for the constructive and helpful comment that helped us to improve the quality of the paper. We add these explanations at the end of the introduction section. The changes are highlighted in yellow and are as follows (lines 103-107):

• In this method, all factors supposed to affect medication errors are included,

• Th103-107e method is based on the principles of fuzzy logic which is the preferred approach in dealing with uncertain and ambiguous data, 

• The method is based on a well-accepted foundation methodology for human error analysis,

• The method is specialized to assess the probability of medication errors,

2-How scalable is the presented approach? We would like the reviewer for this constructive comment. The technique developed in this study is specialized for the assessment of medication error probability. But the methodology is general and can be applied to other domain. But the whole procedure, including identifying subPSFs, weighting them, and rating them must be repeated. 

We add this explanation at the discussion section. The changes are highlighted in yellow and are as follows (lines 315-317): 

The procedure used in this study can be applied to any other domain but all phases, including identifying subPSFs, weighting them, and rating them, must be repeated.

3-This reviewer would like to suggest the authors add a flowchart of the presented method since more details can enrich the paper. We would like to thank the reviewer for this constructive comment. This flowchart was added (Fig 2). 

4-The proposed method might be sensitive to the values of its main controlling parameters. How did you tune the parameters? We would like he reviewer for this comment. After weighting subPSFs and PSFs, the rating process should be performed. In this step, a value within 0-10 is assigned to each parameter (subPSFs). For determining these values, the analyst should visit the workplace (hospital in this case) and interview with the related employees. For more clarity, the following paragraph was added into the case study section (lines 293-296):

Next, the analyst visited the emergency department and interviewed nurses. Based on the direct observation and information obtained from nurses, the rating of subPSFs was determined. Each subPSF can have a value within a range from 1 to 9 so that higher values are indicative of a better situation of that subPSF. SLI and then HEP were calculated based on the rating of subPSFs.

5 The novelty and contribution of the present work need further justification. Key results need to contrast and compare with the results of other state-of-the-art methods in the literature. We would like to thank the reviewer for this constructive comment. We added a paragraph to the discussion section and address this comment. The changes are highlighted in yellow. Moreover, this paragraph are presented below (lines 316-325):

The method developed in this study can evaluate the combined effect of seventeen factors on medication error probability, making it more comprehensive than available HRA techniques. Some of these factors such as workload, availability of work procedures, and safety culture are general to any occupational setting, but some other variables such as patient safety climate, physical health of nurses, and error management culture are unique to healthcare environments. Taking into account these variables is an important advantage of this method over other available techniques. Moreover, the weights calculated in this study are specific to healthcare environments. For example, among organizational factors, patient safety climate had the highest weight, meaning that it is the most important organizational factor affecting the medication error probability. Such a factor with its relative importance has not been considered in any available HRA technique.

Comments for the reviewer #2 and Response of the authors

1 Author should highlight the research gaps and contribution of the proposed work by comparing the state of the art methods and recent studies. First, we would like to thank the reviewer for this constructive comment. This comment helped us to improve the quality of the manuscript considerably. 

We addressed this comment in three section of the manuscript. Two additional paragraphs were added at the end of the introduction section and one paragraph was added to the discussion section. These new paragraphs make it clear why the method presented in this study is the superior one for assessing the medication error probability. All changes are highlighted in yellow. 

Lines 90-100:

Uncertain and ambiguous data and opaque experts’ opinions are other issues in human reliability analysis (HRA). Such data and knowledge can make the results of HRA unreliable. Fuzzy logic theory is specialized for dealing with this issue (26). In contrast to the Boolean logic, the membership degree of a variable to a set can be any real value between zero and one in the Fuzzy environment. Imprecise and vague information obtained in the shape of linguistic terms from domain experts can be manipulated, processed, and represented mathematically using the Fuzzy sets theory and associated Fuzzy operators. Accordingly, Fuzzy logic has found many applications in HRA. For example, techniques such as CREAM, HEART, and SHERPA have been equipped with Fuzzy logic in different ways to reduce subjectivity and enhance the reliability of results (27-30). Consequently, it seems rational, useful, and even necessary to employ Fuzzy logic in developing any new HRA technique.

Lines 103-107:

• In this method, all factors supposed to affect medication errors are included,

• The method is based on the principles of fuzzy logic which is the preferred approach in dealing with uncertain and ambiguous data, 

• The method is based on a well-accepted foundation methodology for human error analysis,

• The method is specialized to assess the probability of medication errors. 

Lines 316-325:

The method developed in this study can evaluate the combined effect of seventeen factors on medication error probability, making it more comprehensive than available HRA techniques. Some of these factors such as workload, availability of work procedures, and safety culture are general to any occupational setting, but some others such as patient safety climate, physical health of nurses, and error management culture are unique to healthcare environments. Taking into account these variables is an important advantage of this method over other available techniques. Moreover, the weights calculated in this study are specific to healthcare environments. For example, among organizational factors, patient safety climate had the highest weight, meaning that it is the most important organizational factor affecting the medication error probability. Such a factor with its relative importance has not been considered in any available HRA technique.

2 Better justify the selected Fuzzy multiple attributive group decision making (MAGDM) methodology and fuzzy analytical hierarchical process (FAHP). There are so many multiple attribute decision making methods. Why did you choose these two methods, based on what evidence? We would like to thank the reviewer for this constructive comment that helped us to improve the quality of the manuscript. We also agree with the reviewer in that there are many FMADM techniques and more justification is needed in the manuscript. Therefore, we added some additional information in the material and methods section. All changes are highlighted in yellow and are as follows:

Line 152-164: 

The main advantage of this method is the use of fuzzy logic in collecting and integrating experts’ opinions. Fuzzy logic is especially helpful in dealing with vague and uncertain information [26]. Accordingly, most multi-criteria decision-making methods have been upgraded based on Fuzzy logic and Fuzzy sets theory [33]. Using Fuzzy logic, the opinions of the experts are gathered in the form of linguistic terms instead of numerical values. Apparently, expressing opinions in terms of linguistic terms is more compatible with the human information processing system and people are more comfortable to express their opinions in terms of linguistic terms instead of crisp values. Moreover, the methodology used in this study benefits from the similarity aggregation method (SAM) for integrating opinions from different experts. SAM is a well-accepted method for making a consensus among opinions gathered from different experts [34]. The method employs a coefficient, known as β, in the aggregation step that provides the analyst or whoever conducts HRA with more dominance and authority over the problem [32]. Moreover, Akyuz [25] used this approach for calculating the weight of PSFs in the SLIM equation.

Lines 220-223:

The reason behind using fuzzy AHP in this phase of the study was its simplicity and also encompassing the advantages of both AHP and Fuzzy logic (37, 38). Moreover, there were only three PSFs in this step, so the number of required pair-wise comparisons was rationally low.

3 In the introduction section, the literature review should be strengthened. Authors are recommended to include and review some studies regarding MAGDM and FAHP applications such as [doi.org/10.1371/journal.pone.0251940], [doi.org/10.1155/2016/3965608], [doi.org/10.1371/journal.pone.0239140] to improve the literature survey: We would like to thank the reviewer for this helpful comment. These references helped us to improve the theoretical basis of the study. 

4 Authors have not presented the limitations of this work. How this work can be extended in practical applications? Please elaborate on it. We would like to thank the reviewer for this constructive comment that helped us to improve the quality of the paper. The limitations of the study was added at the end of the discussion section. 

The added paragraph as below:

Lines 370-374

This study has several limitations which can be the subject of future studies. In this study, the interactions among factors affecting medication errors were not investigated. SLIM cannot capture the interrelationships among a set of PSFs, but other approaches such as Bayesian networks can do this. The weighting process should be repeated in other countries in order to make a comparison about the relative importance of factors affecting medication errors all around the world. 

5 The italic types of symbols in equations and the main text should be in the same expression. We would like the reviewer for this constructive comment. All the expressions were made consistent thorough the manuscript.

6 Although the manuscript is well written in terms of English, there are some (very few, indeed) grammatical and expression errors. It is suggested to proofread the paper. We would like to thank the reviewer for this constructive comment. The manuscript was totally revised in this regard.

---

## [Editor Report · Decision Letter 1]

9 Feb 2022

A comprehensive method for the quantification of medication error probability based on fuzzy SLIM

PONE-D-21-36423R1

Dear Dr. Zahra Pashootan

We’re pleased to inform you that your manuscript has been judged scientifically suitable for publication and will be formally accepted for publication once it meets all outstanding technical requirements.

Kind regards,

Muhammad Junaid Farrukh

Academic Editor

PLOS ONE
---

## [Editor Report · Acceptance letter]

14 Feb 2022

PONE-D-21-36423R1 

A comprehensive method for the quantification of medication error probability based on fuzzy SLIM

Dear Dr. Pashootan:

I'm pleased to inform you that your manuscript has been deemed suitable for publication in PLOS ONE. Congratulations! Your manuscript is now with our production department. 

Kind regards, 

on behalf of

Dr. Muhammad Junaid Farrukh 

Academic Editor

PLOS ONE